green chemistry/chemical engineering

immobilized, ionic liquid, esterification, response surface

**Author for correspondence:**
Ran Liu
e-mail: lrddlr@126.com

This article has been edited by the Royal Society of Chemistry, including the commissioning, peer review process and editorial aspects up to the point of acceptance.

# Catalytic synthesis of *n*-butyl carboxylate with immobilized ionic liquid based on response surface methodology optimization

Ran Liu[1], Ke Zhang[1], Chen Liu[2], Yanhui Hu[1], Lilong Zhou[1] and Juan Zhang[1]

[1]College of Chemistry and Pharmaceutical Engineering, Hebei University of Science and Technology, Shijiazhuang, Hebei 050018, People's Republic of China
[2]College of Medicine and Pharmaceutical Engineering, Shijiazhuang Vocational College of Technology and Information, Shijiazhuang 050018, People's Republic of China

(iD) RL, 0000-0003-3504-257X

Four kinds of functional ionic liquids (ILs) ([$C_3SO_3$Hnmp]$HSO_4$), 1-(3-sulfopropyl)-1-methylpyrrolidone phosphate ([$C_3SO_3$Hnmp]$H_2PO_4$), 1-(3-sulfopropyl)-1-methylpyrrolidone p-toluene sulfonate ([$C_3SO_3$Hnmp]$CH_3SO_3H$) and 1-(3-sulfopropyl)-1-methylpyrrolidone methyl sulfonate ([$C_3SO_3$Hnmp]$C_6H_6SO_3H$)) were prepared and the catalytic activity of these ILs during esterification of carboxylic acids (formic acid, acetic acid, propionic acid, butyric acid) with alcohols was investigated. The results indicated that the IL ([$C_3SO_3$Hnmp]$HSO_4$) exhibited an optimal catalytic performance. And then the IL ([$C_3SO_3$Hnmp]$HSO_4$) was immobilized to the silica gel. The immobilized IL performed more excellent catalytic activity than the unsupported [$C_3SO_3$Hnmp]$HSO_4$. The effects of reaction temperature, reaction time, molar ratio of acid to alcohol and catalyst dosage were investigated. The response surface methodology based on the Box–Behnken design (BBD) was used to explore the best reaction condition of different experimental variables. Accordingly, a high *n*-butyl butyrate yield of 97.10% under the deduced optimal reaction conditions was obtained, in good agreement with experimental results and that predicted by the BBD model. The immobilized IL [$C_3SO_3$Hnmp]$HSO_4$ maintained high catalytic activity after five cycles.

## 1. Introduction

As an important fine organic chemicals product, carboxylic esters have been widely used in perfume, coating, pharmaceutical

intermediates, cosmetics, tobacco and other fields [1]. The main methods to obtain carboxylic esters are extraction of natural substances and chemical synthesis [2,3]. The method of extracting carboxylic ester from natural substance has the disadvantages of complex process, low purity and high production cost. Therefore, the chemical synthesis method was extensively used to prepare carboxylic esters. And the esterification of alcohols and carboxylic acids became fundamental and important reactions in chemical synthesis. Conventionally, chemical syntheses of carboxylic esters mostly invoke homogeneous catalysts, such as sulfuric acid, *p*-toluene sulfonic acid, and phosphoric acid, etc. And using these highly concentrated strong acids as catalyst leads to several problems such as equipment corrosion, environmental pollution, undesirable side reactions and catalyst unrecyclable [4,5]. To avoid these weaknesses, solid super acids, such as heteropoly acids, strong-acid ion exchange resin, zeolites and enzymes, have been employed for esterification [6–9]. However, acid catalysts also have some problems such as low selectivity and catalytic activity, easy deactivation, formidable separation and unrecyclable and so on [10,11].

Ionic liquids (ILs), as new environmental benign catalysts, which were used to replace conventional catalyst, are widely used in various organic synthesis, separation engineering and electrochemistry owing to their unique characteristics such as adjustable properties, designable structure, low melting point, high thermal stability, negligible volatility, recyclability and reusability [12,13]. With further research, the immobilized ILs, which were prepared by grafting IL onto silica gel, molecular sieve, mesoporous nanomaterials and other carriers, can improve the stability and increase catalytic activity in various chemical reactions such as Michael addition, alkylation, epoxidation, Friedle–Crafts and Heck catalytic hydrogenation [14–17]. Herein, we synthesize four kinds of functional ILs (1-(3-sulfopropyl)-1-methylpyrrolidone sulfate ($[C_3SO_3Hnmp]HSO_4$), 1-(3-sulfopropyl)-1-methylpyrrolidone phosphate ($[C_3SO_3Hnmp]H_2PO_4$), 1-(3-sulfopropyl)-1-methylpyrrolidone *p*-toluene sulfonate ($[C_3SO_3Hnmp]CH_3SO_3H$) and 1-(3-sulfopropyl)-1-methylpyrrolidone methyl sulfonate ($[C_3SO_3Hnmp]C_6H_6SO_3H$)) that were used to catalyse the synthesis of *n*-butyl carboxylate. The IL$[C_3SO_3Hnmp]HSO_4$ was immobilized onto silica gel, and the immobilized IL $[C_3SO_3Hnmp]HSO_4$ was applied to the catalytic synthesis of *n*-butyl butyrate for the first time. The relevant reaction conditions obtained during the synthesis of *n*-butyl butyrate were optimized using response surface methodology (RSM). Moreover, the immobilized IL was re-used by a simple separating process, and the catalytic activity of re-used immobilized IL was also investigated.

## 2. Material and methods

### 2.1. Instruments and reagents

*N*-methyl pyrrolidone, 1,3-propane sulfonate lactone, *para*-toluene sulfonic acid, methyl sulfonic acid and tetraethyl orthosilicate were acquired from Aladdin (Shanghai). Methanoic acid, acetic acid, propionic acid, butyric acid and *n*-butyl alcohol were acquired from Tianjin Kemiou Chemical Reagent Co., Ltd. A Fourier transform infrared (FTIR) spectrometer (FTS-135, BIO-RAD) and a gas phase chromatograph (GC-7890II, Tianmei) were used in this study.

### 2.2. Synthesis of ionic liquids

Functionalized ILs, 1-(3-sulfopropyl)-1-methylpyrrolidone sulfate ($[C_3SO_3Hnmp]HSO_4$), 1-(3-sulfopropyl)-1-methylpyrrolidone phosphate ($[C_3SO_3Hnmp]H_2PO_4$), 1-(3-sulfopropyl)-1-methylpyrrolidone *p*-toluene sulfonate ($[C_3SO_3Hnmp]CH_3SO_3H$) and 1-(3-sulfopropyl)-1-methylpyrrolidone methyl sulfonate ($[C_3SO_3Hnmp]C_6H_6SO_3H$), were synthesized. The supported IL $[C_3SO_3Hnmp]H_2PO_4$ was synthesized according to previous studies [18–20]. The supported IL $[C_3SO_3Hnmp]HSO_4$ was synthesized according to previous studies [21,22].

### 2.3. Synthesis of esters

A certain amount of formic acid, *n*-butanol, catalysts (ILs, supported ILs) and water as a carrying agent were proportionally added to three-necked flasks with reflux condensers, water distributors and magnetic stirring. These three-necked flasks were placed in an oil bath for 4 h at 110°C. The mixture obtained by the reaction was evaporated by a vacuum rotary evaporator at 85°C. After the reaction, the catalyst was recovered, dried and re-used, and its reusability was measured. The rotary evaporation solution products were weighed. The crude esters were washed with saturated sodium chloride and saturated sodium carbonate solutions. Unreacted formic acid and *n*-butanol were

**Table 1.** Solubility of catalysts in the reaction system. s, solution; i, insoluble.

| ionic liquid | water | acid | alcohol | ester products |
|---|---|---|---|---|
| $[C_3SO_3Hnmp]HSO_4$ | s | i | s | i |
| $[C_3SO_3Hnmp]H_2PO_4$ | s | i | s | i |
| $[C_3SO_3Hnmp]CH_3SO_3H$ | s | i | s | i |
| $[C_3SO_3Hnmp]C_6H_6SO_3H$ | s | i | s | i |
| Immobilized $[C_3SO_3Hnmp]HSO_4$ | i | i | i | i |

**Table 2.** Catalytic activity of ILs.

| ionic liquid | N-butyl formate (%) | N-butyl acetate (%) | N-butyl propionate (%) | N-butyl butyrate (%) |
|---|---|---|---|---|
| $[C_3SO_3Hnmp]HSO_4$ | 88.29 | 86.33 | 89.21 | 89.89 |
| $[C_3SO_3Hnmp]H_2PO_4$ | 80.36 | 86.67 | 88.57 | 88.32 |
| $[C_3SO_3Hnmp]CH_3SO_3H$ | 87.21 | 85.20 | 85.36 | 87.21 |
| $[C_3SO_3Hnmp]C_6H_6SO_3H$ | 80.73 | 82.99 | 83.32 | 83.54 |

removed. Then, the target product, *n*-butyl formate, was obtained by drying with anhydrous magnesium sulfate. The yield of *n*-butyl formate was calculated as a proportion. This method also led to the production of *n*-butyl acetate, *n*-butyl propionate and *n*-butyl butyrate.

# 3. Results and discussion

## 3.1. Solubility of catalysts

The solubility of four ILs and supported ILs $[C_3SO_3Hnmp]HSO_4$ with water, acids, alcohols and esters was investigated. As shown in table 1, the four ILs were insoluble in acids and their products and were soluble in water and alcohols. After the reaction, catalysts and products of the ILs were automatically stratified and easily separated. The supported ILs $[C_3SO_3Hnmp]HSO_4$ were insoluble in all reactants and products. Therefore, this approach has more convenient operation conditions and promotes the reaction.

## 3.2. Catalytic activity of ionic liquids

Four kinds of ILs were used as catalysts to catalyse esterification under the following conditions: a reaction time of 4 h, a reaction temperature of 110°C, an acid–alcohol ratio of 1.2 : 1 and a catalyst dosage 5%. After the reaction, the esters were washed with saturated sodium carbonate and saturated sodium chloride solutions then dried with anhydrous magnesium sulfate. The product was placed in a vacuum drying chamber for 1.5 h, and then the product yield was measured as shown in table 2.

Table 2 shows that IL $[C_3SO_3Hnmp]HSO_4$ has the strongest catalytic activity and the highest yield of *n*-butyl butyrate. Because the catalytic activity of acidic ILs is related to the strength of anionic acidity and the anion $HSO_4^-$ of IL $[C_3SO_3Hnmp]HSO_4$ has relatively strong acidity, this IL shows the highest catalytic activity in esterification.

## 3.3. Study of the technological conditions of esterification catalysed by supported ionic liquid $[C_3SO_3Hnmp]HSO_4$

Catalytic synthesis of *n*-butyl butyrate was performed using the supported IL $[C_3SO_3Hnmp]HSO_4$ as a catalyst. The effects of reaction temperature, reaction time, acid–alcohol ratio and catalyst dosage on the catalytic synthesis of *n*-butyl butyrate were studied.

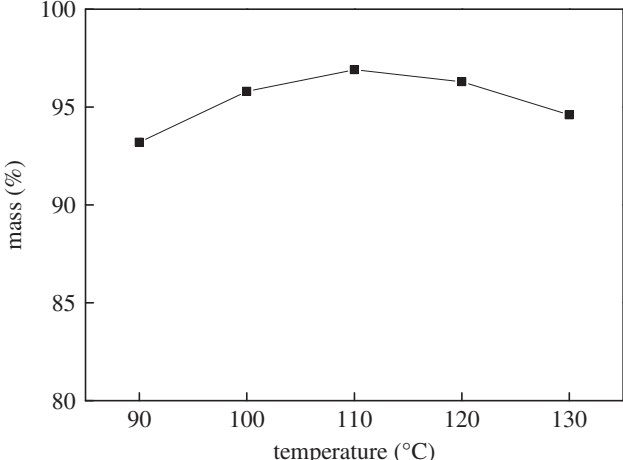

**Figure 1.** Effect of reaction temperature on the yield of *n*-butyl butyrate.

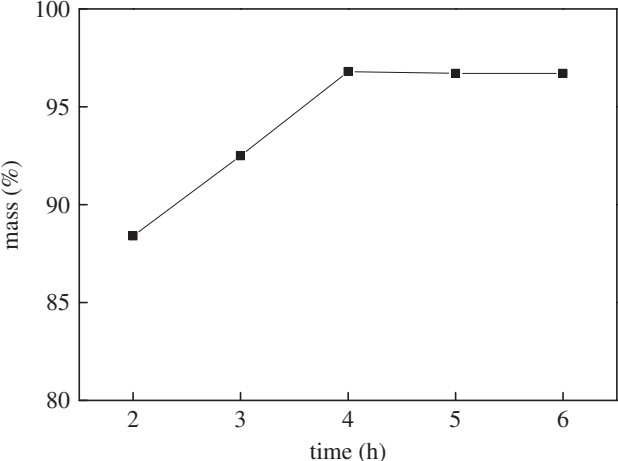

**Figure 2.** Effect of reaction time on the yield of *n*-butyl butyrate.

### 3.3.1. Effect of reaction temperature on the yield of *n*-butyl butyrate

The reaction temperatures were controlled at 90, 100, 110, 120 and 130°C. After the reaction, the mass of *n*-butyl butyrate was measured and the yield was calculated. The results are shown in figure 1.

Figure 1 shows that the yield of *n*-butyl butyrate first increases and then decreases slightly with increasing reaction temperature. The yield of *n*-butyl butyrate increased with increasing temperature. However, the reaction is exothermic, and excessive temperature will lead to an increase in the reverse reaction, and furthermore, the activity of the catalyst may be affected by high temperature, both of which would decrease the yield of *n*-butyl butyrate. Therefore, the optimum reaction temperature was found to be 110°C.

### 3.3.2. Effect of reaction time on the yield of *n*-butyl butyrate

The reaction time was controlled for 2, 3, 4, 5 and 6 h at 110°C. After the reaction, the mass of *n*-butyl butyrate was measured, and the yield was calculated. The results are shown in figure 2.

Figure 2 shows that the yield of *n*-butyl butyrate increases with increasing reaction time. The yield of *n*-butyl butyrate remains unchanged after 4 h, as the reaction is complete after 4 h; therefore, the optimal reaction time is 4 h.

### 3.3.3. Effect of acid–alcohol ratio on the yield of *n*-butyl butyrate

With a reactor temperature of 110°C, a reaction time of 4 h and a catalyst amount of 5%, the ratio of acid to alcohol was examined at 1 : 1, 1.1 : 1, 1.2 : 1, 1.3 : 1 and 1.4 : 1. After the reaction, the quality of *n*-butyl butyrate was measured, and the yield was calculated, as shown in figure 3.

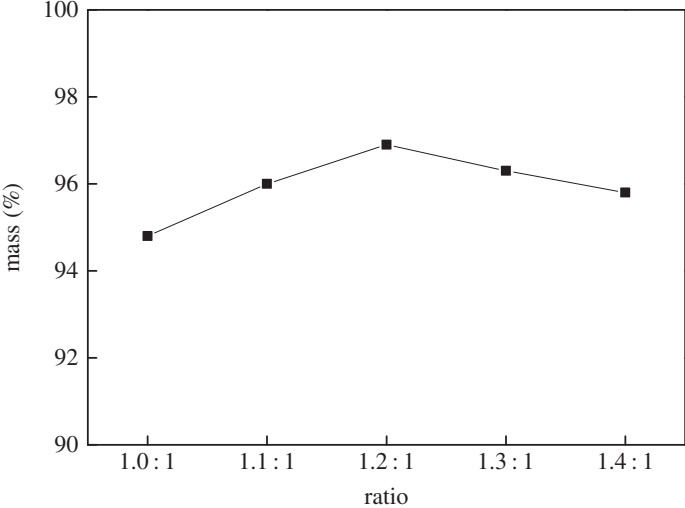

**Figure 3.** Effect of acid and alcohol ratio on the yield of n-butyl butyrate.

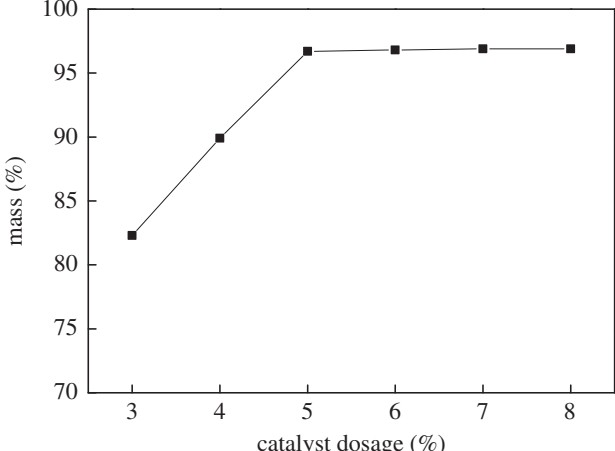

**Figure 4.** Effect of [C$_3$SO$_3$Hnmp]HSO$_4$ dosage on the yield of n-butyl butyrate.

Figure 3 shows that the yield of n-butyl butyrate first increases and then decreases slightly with the increasing acid–alcohol ratio. As the amount of acid increases gradually, the yield of n-butyl butyrate increases. When the ratio of acid to alcohol exceeds 1.2 : 1, the yield of n-butyl butyrate decreases slightly, we speculated that this is due to the excessive amount of butyric acid, the introduction of excessive water and the reduced concentration of the reaction solution.

### 3.3.4. Effect of the dosage of the supported ionic liquid [C$_3$SO$_3$Hnmp]HSO$_4$ on the yield of n-butyl butyrate

With a reactor temperature of 110°C, a reaction time of 4 h and a ratio of acid to alcohol of 1.2 : 1, the amount of catalyst used was 3, 4, 5, 6, 7 and 8% of the amount of n-butanol. After the reaction, the mass of n-butyl butyrate was measured, and the yield was calculated. The results are shown in figure 4.

As shown in figure 4, the yield of n-butyl butyrate increased rapidly from 83 to 96.8% with the increasing amount of supported IL [C$_3$SO$_3$Hnmp]HSO$_4$. When the amount of catalyst exceeds 5%, the yield of n-butyl butyrate is less than 1%. Therefore, the amount of catalyst used is 5%.

## 3.4. Optimization of RSM process conditions to synthesize n-butyl butyrate

Under the single-factor optimum conditions, a molar ratio of acid to alcohol of 1.2 : 1, a reaction temperature of 110°C and a catalyst dosage of 5% of n-butanol, RSM [23] was used to optimize the experiment.

**Table 3.** The coding level of each factor in the test design.

| factor | variable | level | | |
|---|---|---|---|---|
| | | −1 | 0 | 1 |
| $t$ (h) | $A$ | 3 | 4 | 5 |
| mole ratio | $B$ | 1.0 | 1.2 | 1.4 |
| $T$ (°C) | $C$ | 100 | 110 | 120 |
| $\omega$(DES) (%) | $D$ | 3 | 5 | 7 |

### 3.4.1. Design and result of response surface experiment scheme

The reaction time, molar ratio of butyric acid to $n$-butanol, reaction temperature and catalyst dosage were marked as influencing factors $A$, $B$, $C$ and $D$, respectively. The yield of $n$-butyl butyrate was used as the response value to optimize the experimental conditions using a Box–Behnken model in Design-Expert software. The graph was used to show the functional relationships of various factors, and then the optimal conditions for the experimental design were determined [24]. Experiments were divided into 29 groups, in which experiments 1–24 were factorial experiments to test whether there was interaction between various factors, and experiments 25–29 were central experiments to calculate experimental errors and evaluate them. The coding level of each factor in the experimental design is shown in table 3, and the experimental scheme and response surface analysis results are shown in table 4.

### 3.4.2. Establishment of the model and significance test

The results of optimization are shown in table 5, with $F = 14.74$ and $p < 0.0001$, which indicates that the quadratic regression model is more significant. The linear relationship between response value $Y$ and influencing factors $A$, $B$, $C$ and $D$ is obvious. Therefore, the regression model can be used to predict the yield of $n$-butyl butyrate. According to the analysis of the $F$-value in table 5, the reaction time has the greatest influence on the response value followed by the amount of catalyst, reaction temperature and molar ratio of acid to alcohol. The interaction of the four factors has a great influence on the yield of $n$-butyl butyrate. The experimental data of the yield $Y$ of $n$-butyl butyrate showed multivariate regression. Finally, the quadratic polynomial regression equation of the yield of $n$-butyl butyrate $Y$ for each influencing factor was obtained. Ultimately, the quadratic polynomial regression equation for the yield of $n$-butyl butyrate $Y$ for each influencing factor was determined to be $Y = +88.63 + 0.032 \times A - 0.85 \times B + 0.98 \times C - 1.41 \times D - 1.72 \times A \times B - 0.44 \times A \times C + 1.21 \times A \times D + 2.91 \times B \times C - 2.69 \times B \times D - 0.34 \times C \times D - 3.37 \times A^2 - 2.30 \times B^2 + 3.23 \times C^2 - 0.34 \times D^2$.

### 3.4.3. Interaction among various factors

Interactions exist among reaction time ($A$), molar ratio of acid to alcohol ($B$), reaction temperature ($C$) and catalyst dosage ($D$). Response surface analysis of the interaction between the four factors was performed, and contours were drawn as shown in figures 5–10. (R1 = yield.)

Figure 5 shows that the yield of $n$-butyl butyrate increases with increasing catalyst dosage when the molar ratio of acid to alcohol is constant. When the amount of catalyst is constant, the yield of $n$-butyl butyrate increases first and then decreases with increasing amount of butyric acid. It can be seen from the contour that the interaction between the molar ratio of acid to alcohol and the amount of catalyst is significant, which is consistent with the variance analysis results.

Figure 6 shows that the yield of $n$-butyl butyrate increases with increasing reaction temperature when the molar ratio of acid to alcohol is constant, and increases first and then decreases with increasing molar ratio of acid to alcohol when the reaction temperature is constant. Compared with the molar ratio of acid to alcohol, the effect of reaction temperature on the yield of $n$-butyl butyrate is more significant. From the contour, it can be seen that the results are consistent with the variance analysis and the interaction between the molar ratio of acid to alcohol and reaction temperature is not significant.

Figure 7 shows that the yield of $n$-butyl butyrate is positively correlated with reaction time and temperature by single-factor experiments. From the contour map, we can see that the interaction between them is obvious, which is consistent with variance analysis results.

**Table 4.** Response surface analysis results.

| std | level | | | | Y (%) | std | level | | | | Y (%) |
| | A | B | C | D | | | A | B | C | D | |
|---|---|---|---|---|---|---|---|---|---|---|---|
| 1 | −1 | −1 | 0 | 0 | 94.38 | 16 | 0 | 1 | 1 | 0 | 96.23 |
| 2 | 1 | −1 | 0 | 0 | 96.23 | 17 | −1 | 0 | −1 | 0 | 96.56 |
| 3 | −1 | 1 | 0 | 0 | 96.96 | 18 | 1 | 0 | −1 | 0 | 96.01 |
| 4 | 1 | 1 | 0 | 0 | 91.93 | 19 | −1 | 0 | 1 | 0 | 96.48 |
| 5 | 0 | 0 | −1 | −1 | 93.23 | 20 | 1 | 0 | 1 | 0 | 94.19 |
| 6 | 0 | 0 | 1 | −1 | 96.36 | 21 | 0 | −1 | 0 | −1 | 90.15 |
| 7 | 0 | 0 | −1 | 1 | 89.84 | 22 | 0 | 1 | 0 | −1 | 95.08 |
| 8 | 0 | 0 | 1 | 1 | 91.63 | 23 | 0 | −1 | 0 | 1 | 94 |
| 9 | −1 | 0 | 0 | −1 | 92.22 | 24 | 0 | 1 | 0 | 1 | 88.16 |
| 10 | 1 | 0 | 0 | −1 | 93.01 | 25 | 0 | 0 | 0 | 0 | 93.52 |
| 11 | −1 | 0 | 0 | 1 | 86.96 | 26 | 0 | 0 | 0 | 0 | 93.19 |
| 12 | 1 | 0 | 0 | 1 | 92.58 | 27 | 0 | 0 | 0 | 0 | 83.19 |
| 13 | 0 | −1 | −1 | 0 | 95.62 | 28 | 0 | 0 | 0 | 0 | 85.87 |
| 14 | 0 | 1 | −1 | 0 | 86.03 | 29 | 0 | 0 | 0 | 0 | 87.39 |
| 15 | 0 | −1 | 1 | 0 | 94.17 | | | | | | |

**Table 5.** Variance analysis table.

| source | sum of squares | d.f. | mean square | F-value | p-value | |
|---|---|---|---|---|---|---|
| model | 300.61 | 14 | 21.47 | 14.74 | <0.0001 | significant |
| A—t | 45.67 | 1 | 45.67 | 31.35 | <0.0001 | |
| B—mole ratio | 8.60 | 1 | 8.60 | 0.87 | 0.3666 | |
| C—T | 11.54 | 1 | 11.54 | 1.17 | 0.2980 | |
| D—W | 23.74 | 1 | 23.74 | 2.40 | 0.1434 | |
| AB | 11.83 | 1 | 11.83 | 1.20 | 0.2923 | |
| AC | 0.76 | 1 | 0.76 | 0.077 | 0.7860 | |
| AD | 5.83 | 1 | 5.83 | 0.59 | 0.4551 | |
| BC | 33.93 | 1 | 33.93 | 3.43 | 0.0851 | |
| BD | 29.00 | 1 | 29.00 | 2.93 | 0.1087 | |
| CD | 0.45 | 1 | 0.45 | 0.045 | 0.8343 | |
| A2 | 73.77 | 1 | 73.77 | 7.47 | 0.0162 | |
| B2 | 34.35 | 1 | 34.35 | 3.48 | 0.0834 | |
| C2 | 67.56 | 1 | 67.56 | 6.84 | 0.0204 | |
| D2 | 0.73 | 1 | 0.73 | 0.074 | 0.7894 | |
| residual | 138.33 | 14 | 9.88 | | | |
| lack of fit | 54.88 | 10 | 5.49 | 0.26 | 0.9605 | not significant |
| pure error | 83.45 | 4 | 20.86 | | | |
| cor total | 399.23 | 28 | | | | |

Figure 8 shows that the yield of $n$-butyl butyrate increases with increasing amount of catalyst when the reaction time is fixed and increases the reaction time when the amount of catalyst is fixed. Compared with the amount of catalyst, the effect of reaction time on the yield of $n$-butyl butyrate is

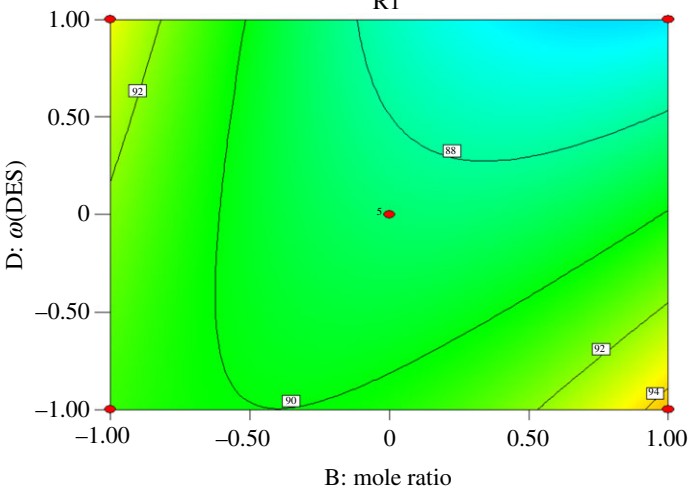

**Figure 5.** Effect of *B* and *D* on the yield of *n*-butyl butyrate.

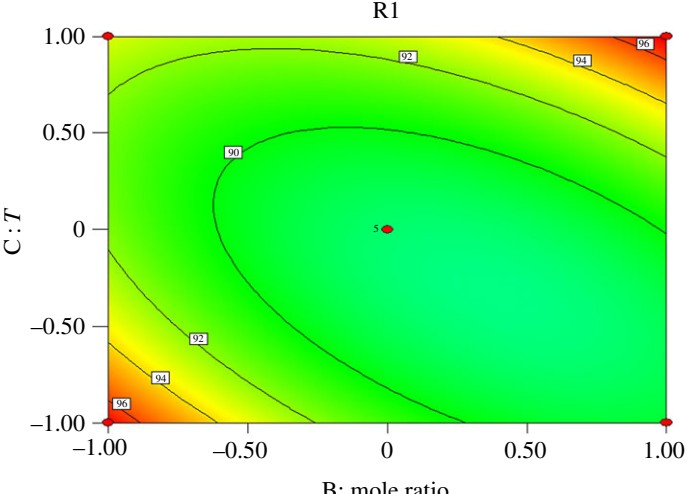

**Figure 6.** Effect of *B* and *C* on the yield of *n*-butyl butyrate.

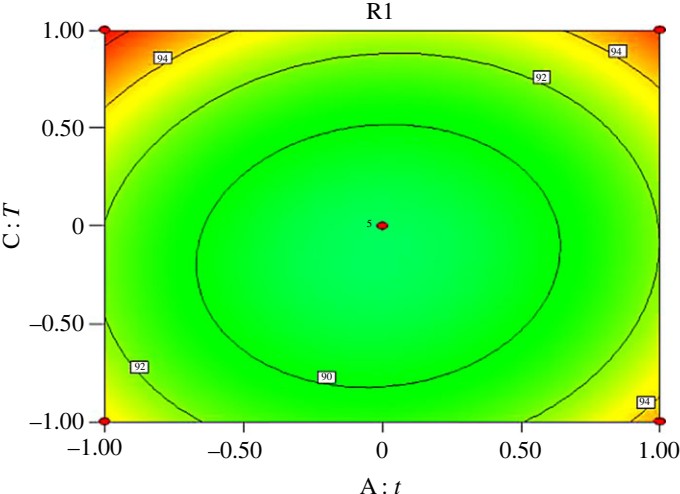

**Figure 7.** Effect of *A* and *C* on the yield of *n*-butyl butyrate.

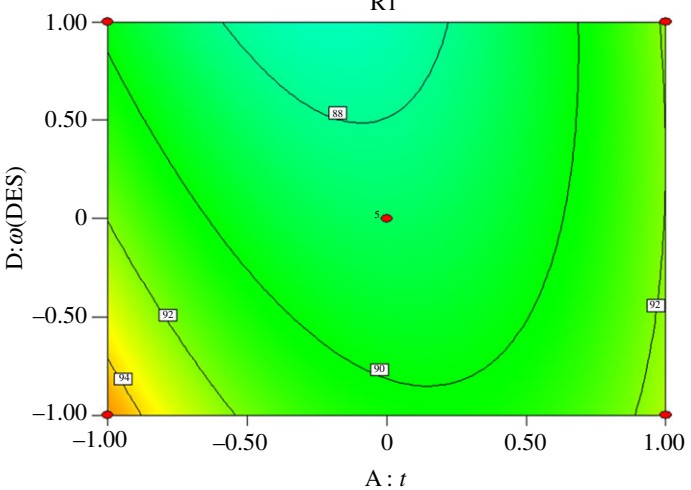

**Figure 8.** Effect of *A* and *D* on the yield of *n*-butyl butyrate.

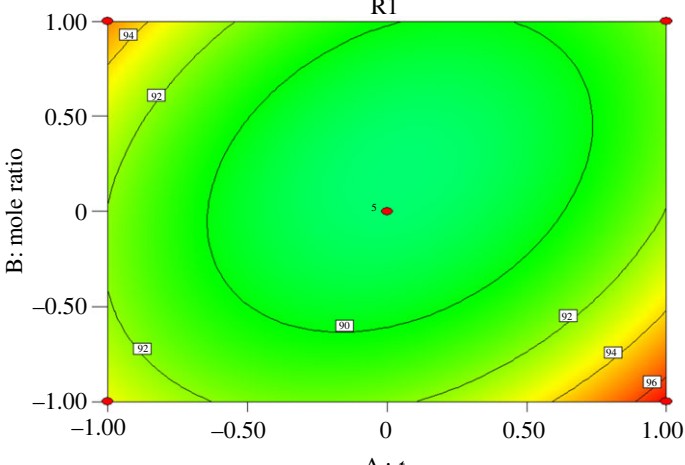

**Figure 9.** Effect of *A* and *B* on the yield of *n*-butyl butyrate.

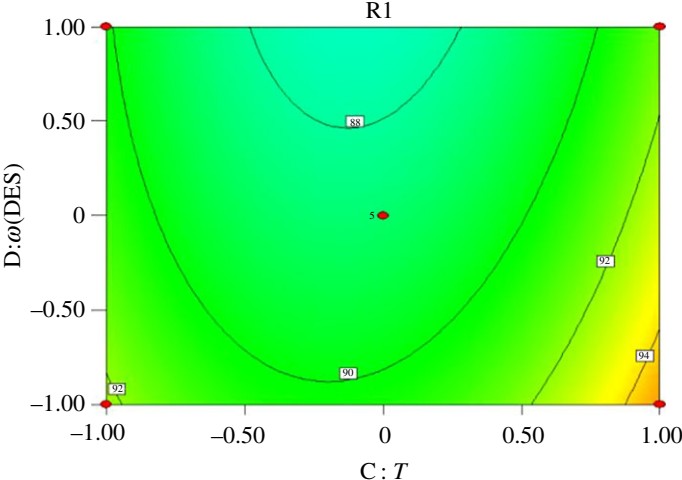

**Figure 10.** Effect of *C* and *D* on the yield of *n*-butyl butyrate.

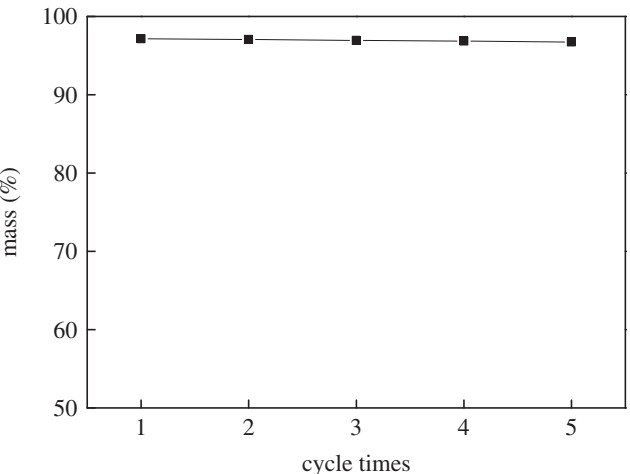

**Figure 11.** Reuse performance of the immobilized IL [C$_3$SO$_3$Hnmp]HSO$_4$.

more significant, and the interaction between the two is significant, which is consistent with the variance analysis results.

Figure 9 shows that the yield of *n*-butyl butyrate increases first and then decreases with increasing amount of butyric acid at a given reaction time. Increasing the amount of butyric acid will make the esterification reaction move to the right. However, an excessive amount of butyric acid will reduce the concentration of *n*-butanol and catalyst, which is not conducive to the positive reaction and will reduce the yield of *n*-butyl butyrate when the molar ratio of acid to alcohol is fixed. The yield of *n*-butyl butyrate increased with reaction time. The reaction time and molar ratio of acid to alcohol had significant effects on the yield of *n*-butyl butyrate, but the interaction between them was not significant.

Figure 10 shows that the yield of *n*-butyl butyrate is positively correlated with reaction temperature and catalyst dosage by single-factor experiments. The contour plot was round, indicating that the interaction between reaction temperature and catalyst dosage was not significant, which was consistent with the results of variance analysis.

### 3.4.4. Determination of optimal reaction conditions

Based on response surface and quadratic regression model analysis combined with the influence of various factors on the yield of *n*-butyl butyrate, the optimum conditions for esterification with supported IL [C$_3$SO$_3$Hnmp]HSO$_4$ as catalyst were obtained as follows: a reaction time of 4 h, a molar ratio of acid to alcohol of 1.14 : 1, a reaction temperature of 120°C, a catalyst dosage 7% of the butanol dosage and a predicted yield of *n*-butyl butyrate of 97.21%. Under the above optimum conditions, three experiments were performed with the supported IL [C$_3$SO$_3$Hnmp]HSO$_4$ as catalyst. The yields of *n*-butyl butyrate were 97.12, 97.10 and 97.14%, which were close to the predicted yield; therefore, the optimum conditions were reliable.

### 3.5. Reuse performance of supported ionic liquid [C$_3$SO$_3$Hnmp]HSO$_4$

The immobilized IL [C$_3$SO$_3$Hnmp]HSO$_4$ can be re-used after recovery and drying. Figure 11 shows that under the optimum reaction conditions, the yield of *n*-butyl butyrate synthesized by supported IL [C$_3$SO$_3$Hnmp]HSO$_4$ is above 96% after five cycles, and its catalytic performance is good.

### 3.6. Study on the mechanism of ester synthesis catalysed by the supported ionic liquid [C$_3$SO$_3$Hnmp]HSO$_4$

The supported IL [C$_3$SO$_3$Hnmp]HSO$_4$ can provide a proton, H$^+$, which is similar to the mechanism of proton acidic catalytic esterification. It is speculated that the reaction process provides H$^+$ for [C$_3$SO$_3$Hnmp]HSO$_4$ and combines with carbon atoms on the carboxyl group of the C$_3$H$_7$COOH molecule to cause the carbon atoms on the carboxyl group to have stronger electrophilicity, thereby causing the oxygen on the alcohol hydroxyl group to react more easily. Then, H$^+$ combines with the

**Figure 12.** Esterification mechanism catalysed by immobilized [C₃SO₃Hnmp]HSO₄.

carboxyl oxygen in the intermediate, the hydroxyl is protonated to remove a water molecule, and one proton is lost from the second hydroxyl to form *n*-butyl butyrate [25]. The reaction mechanism is shown in figure 12.

# 4. Conclusion

In this paper, a series of Brønsted acidic ILs were synthesized and exploited to catalyse esterification of butyric acid and butyl alcohol. The IL [C₃SO₃Hnmp]HSO₄ has the best catalytic activity and the highest yield of *n*-butyl butyrate. So, the IL [C₃SO₃Hnmp]HSO₄ was selected to be immobilized using the sol–gel method. With the optimal esterification conditions: a molar ratio of acid to alcohol of 1.2 : 1, a reaction temperature of 110°C and a catalyst dosage of 5% of *n*-butanol, a maximum *n*-butyl butyrate yield of 97.10% was obtained catalysed by the immobilized IL [C₃SO₃Hnmp]HSO₄. This result is consistent with the prediction of the response surface optimization based on the Box–Behnken design model. Using immobilized IL [C₃SO₃Hnmp]HSO₄ as catalyst can avoid the mutual dissolution of the catalyst and the solvent, and provide a larger specific surface area, which leads to a significant increase in the yield of *n*-butyl butyrate. The yield of *n*-butyl butyrate remained above 96% after re-using the supported IL [C₃SO₃Hnmp]HSO₄ five times, demonstrating good reusability.

Data accessibility. The datasets supporting this article have been uploaded as part of the electronic supplementary material.

Authors' contributions. R.L. and K.Z. carried out the laboratory work, conceived of the study, participated in data analysis, participated in the design of the study and drafted the manuscript; C.L. and Y.H. designed the study, carried out the statistical analyses and critically revised the manuscript; L.Z. and J.Z. collected field data and critically revised the manuscript; all authors gave final approval for publication and agree to be held accountable for the work performed therein.

Competing interests. The authors declare no competing interests.

Funding. This work was supported by the National Natural Science Foundation of China (no. 21106032), scientific treatment of haze and air pollution prevention and control key research projects of Hebei University of Science and Technology (no. 2017020) and College Students Innovation and Entrepreneurship Plan Training Program (2018017, 201910082020).

Acknowledgements. The authors thank Prof. Dishun Zhao for help on FTIR and GC measurements.

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
