## [Reviewer comments · Royal Society Open Science]

Review History

RSOS-190166.R0 (Original submission)

Review form: Reviewer 1

Is the manuscript scientifically sound in its present form?

Yes

Are the interpretations and conclusions justified by the results?

Yes

Is the language acceptable?

No

Is it clear how to access all supporting data?

Yes

Do you have any ethical concerns with this paper?

No

Have you any concerns about statistical analyses in this paper?

No

Recommendation?

Major revision is needed (please make suggestions in comments)

Comments to the Author(s)

Thanks for inviting me to review this paper. The work is interesting and merit publication in the journal. However, the English is poor and need to be polished. The following points are given for improvement:

Page 1, lines 27-31: The sentence is very difficult to be understood. Which is the subject of the article needs to be clearly explained. Too tedious in current form.

Page 1, lines 42-43: please replace "studied" for "were studied".

Page 2, lines 23: please replace "three flasks" to "three-necked flask". Similar mistakes should be carefully checked.

Page 2, lines 21-30: How do you determine the purity of the products? The differences of the yields were not large (83%-96.8%), but the accuracy of the yield has a great influence on the establishment of the model.

Page 8, lines 46-50: Is any catalyst weight loss detected during the recycling processes?

Review form: Reviewer 2

Is the manuscript scientifically sound in its present form?

No

Are the interpretations and conclusions justified by the results?

Yes

Is the language acceptable?

Yes

Is it clear how to access all supporting data?

Yes

Do you have any ethical concerns with this paper?

Yes

Have you any concerns about statistical analyses in this paper?

No

Recommendation?

Major revision is needed (please make suggestions in comments)

Comments to the Author(s)

The manuscript (ID: RSOS-190166), titled "Catalytic Synthesis of N-Butyl Carboxylate with Immobilized Ionic Liquid Based on RSM Optimization" details a catalytic methodology towards the synthesis of several esters. The reaction scope was evaluated as well as several other aspects such as reaction temperature, reaction time, reutilization of the catalyst system, etc... The results are interesting and probably will interest several researches. However, in my opinion the authors

should rewrite the abstract, introduction and conclusions because I detect plagiarism in those parts.

I do think that experimental section and/or discussion can have similar descriptions but in the other parts the authors should be innovative.

Decision letter (RSOS-190166.R0)

01-May-2019

Dear Dr Liu:

Title: Catalytic Synthesis of N-Butyl Carboxylate with Immobilized Ionic Liquid Based on RSM Optimization

Manuscript ID: RSOS-190166

The editor assigned to your manuscript has now received comments from reviewers. We would like you to revise your paper in accordance with the referee and Subject Editor suggestions which can be found below (not including confidential reports to the Editor). Please note this decision does not guarantee eventual acceptance. I apologise that this has taken longer than usual.

Please submit your revised paper before 24-May-2019. Please note that the revision deadline will expire at 00.00am on this date. If we do not hear from you within this time then it will be assumed that the paper has been withdrawn. In exceptional circumstances, extensions may be possible if agreed with the Editorial Office in advance. We do not allow multiple rounds of revision so we urge you to make every effort to fully address all of the comments at this stage. If deemed necessary by the Editors, your manuscript will be sent back to one or more of the original reviewers for assessment. If the original reviewers are not available we may invite new reviewers.

On behalf of the Subject Editor Professor Anthony Stace and the Associate Editor Professor John Moses.

RSC Associate Editor:

Comments to the Author:

Reviewer 2 has suggested that there is possible text overlap in the manuscript with other published works. Both the Royal Society and the Royal Society of Chemistry are members of COPE (Committee on Publication Ethics) and our ethical publishing policies align with their best practice standards.

We therefore contacted the reviewer and they have provided additional comments as follows: "In this manuscript I think the authors should rewrite the introduction, in particularly the first part " Esterification, one of the most important reactions in organic synthesis, is widely used in perfumes, plasticizers, solvents and other chemical products [1]. With economic development; stricter requirements have been imposed on the industrial production of ester products. The traditional esterification reaction using concentrated sulfuric acid as strong acid catalyst leads to such issues as strong corrosiveness, difficult treatment of waste liquor, and many side reactions [2-3]. Many kinds of heteropoly acids studied later as catalysts also result in such problems as low catalytic efficiency, volatility, and difficulty in recovery [4]. However, the traditional supported acid catalysts have such problems as low selectivity, low catalytic activity and harsh reaction conditions [5]. In recent years, functionalized ionic liquids as new solvents and catalysts have met the requirements of "green chemistry" and are easy to recover. These ionic liquids are widely used in catalytic reactions, separation engineering and electrochemistry, and because of their unique catalytic properties, they have attracted increasing attention [6]."

We ask you to consider re-wording the abstract, introduction and conclusions to take this into account.

RSC Subject Editor:

Comments to the Author:

(There are no comments.)

Reviewers' Comments to Author:

Reviewer: 1

Comments to the Author(s)

Thanks for inviting me to review this paper. The work is interesting and merit publication in the journal. However, the English is poor and need to be polished. The following points are given for improvement:

Page 1, lines 27-31: The sentence is very difficult to be understood. Which is the subject of the article needs to be clearly explained. Too tedious in current form.

Page 1, lines 42-43: please replace "studied" for "were studied".

Page 2, lines 23: please replace “three flasks” to “three-necked flask”. Similar mistakes should be carefully checked.

Page 2, lines 21-30: How do you determine the purity of the products? The differences of the yields were not large (83%-96.8%), but the accuracy of the yield has a great influence on the establishment of the model.

Page 8, lines 46-50: Is any catalyst weight loss detected during the recycling processes?

Reviewer: 2

Comments to the Author(s)

The manuscript (ID: RSOS-190166), titled “Catalytic Synthesis of N-Butyl Carboxylate with Immobilized Ionic Liquid Based on RSM Optimization” details a catalytic methodology towards the synthesis of several esters. The reaction scope was evaluated as well as several other aspects such as reaction temperature, reaction time, reutilization of the catalyst system, etc... The results are interesting and probably will interest several researches. However, in my opinion the authors should rewrite the abstract, introduction and conclusions because I detect plagiarism in those parts.

I do think that experimental section and/or discussion can have similar descriptions but in the other parts the authors should be innovative.

Author's Response to Decision Letter for (RSOS-190166.R0)

See Appendix A.

RSOS-190166.R1 (Revision)

Review form: Reviewer 1

Is the manuscript scientifically sound in its present form?

Yes

Are the interpretations and conclusions justified by the results?

Yes

Is the language acceptable?

Yes

Do you have any ethical concerns with this paper?

No

Have you any concerns about statistical analyses in this paper?

No

Recommendation?

Accept as is

Comments to the Author(s)

The manuscript has been improved decently to reach the requirement of the journal. Therefore, I recommend publication of this manuscript at present stage.

Review form: Reviewer 2

Is the manuscript scientifically sound in its present form?

Yes

Are the interpretations and conclusions justified by the results?

Yes

Is the language acceptable?

Yes

Do you have any ethical concerns with this paper?

No

Have you any concerns about statistical analyses in this paper?

No

Recommendation?

Accept with minor revision (please list in comments)

Comments to the Author(s)

The authors improved the manuscript and it is much better now. However, they should decide if the figures will be presented in the manuscript or in the supplementary data. In my opinion they should put less in the manuscript. Nevertheless, they should not repeat the same figures in both documents.

Decision letter (RSOS-190166.R1)

01-Jul-2019

Dear Dr Liu:

Title: Catalytic Synthesis of N-Butyl Carboxylate with Immobilized Ionic Liquid Based on RSM Optimization

Manuscript ID: RSOS-190166.R1

Thank you for submitting the above manuscript to Royal Society Open Science. On behalf of the Editors and the Royal Society of Chemistry, I am pleased to inform you that your manuscript will be accepted for publication in Royal Society Open Science subject to minor revision in accordance with the referee suggestions. Please find the reviewers' comments at the end of this email.

The reviewers and handling editors have recommended publication, but also suggest some minor revisions to your manuscript. Therefore, I invite you to respond to the comments and revise your manuscript.

Because the schedule for publication is very tight, it is a condition of publication that you submit the revised version of your manuscript before 10-Jul-2019. Please note that the revision deadline will expire at 00.00am on this date. If you do not think you will be able to meet this date please let me know immediately.

Best wishes,
Dr Laura Smith
Publishing Editor, Journals

Royal Society of Chemistry
Thomas Graham House

Science Park, Milton Road
Cambridge, CB4 0WF
Royal Society Open Science - Chemistry Editorial Office

RSC Associate Editor:
Comments to the Author:
Please respond to the comments about figures from Reviewer 2.

RSC Subject Editor:
Comments to the Author:
(There are no comments.)

Reviewer comments to Author:
Reviewer: 1

Comments to the Author(s)
The manuscript has been improved decently to reach the requirement of the journal. Therefore, I recommend publication of this manuscript at present stage.

Reviewer: 2

Comments to the Author(s)
The authors improved the manuscript and it is much better now. However, they should decide if the figures will be presented in the manuscript or in the supplementary data. In my opinion they should put less in the manuscript. Nevertheless, they should not repeat the same figures in both documents.

Author's Response to Decision Letter for (RSOS-190166.R1)

See Appendix B.

Decision letter (RSOS-190166.R2)

22-Jul-2019

Dear Dr Liu:

Title: Catalytic Synthesis of N-Butyl Carboxylate with Immobilized Ionic Liquid Based on RSM Optimization
Manuscript ID: RSOS-190166.R2

It is a pleasure to accept your manuscript in its current form for publication in Royal Society Open Science. The chemistry content of Royal Society Open Science is published in collaboration with the Royal Society of Chemistry.

RSC Associate Editor
Comments to the Author:
(There are no comments.)

Reviewer(s)' Comments to Author:

Appendix A

Response to comments of editor and reviewers

Dear editor and reviewers:

Thank you for your comments concerning our manuscript entitled “Catalytic Synthesis of N-Butyl Carboxylate with Immobilized Ionic Liquid Based on RSM” (ID: RSOS-190166), Those comments are all valuable and very helpful for revising and improving our paper, as well as the important guiding significance to our researches. We have studied comments carefully and have made correction which we hope meet with approval. Revised portion are marked in red in the paper. The main corrections in the paper and the responds to comments are as flowing:

Response to comments of RSC Associate Editor:

Reviewer 2 has suggested that there is possible text overlap in the manuscript with other published works. Both the Royal Society and the Royal Society of Chemistry are members of COPE (Committee on Publication Ethics) and our ethical publishing policies align with their best practice standards.

We therefore contacted the reviewer and they have provided additional comments as follows: "In this manuscript I think the authors should rewrite the introduction, in particularly the first part " Esterification, one of the most important reactions in organic synthesis, is widely used in perfumes, plasticizers, solvents and other chemical products [1]. With economic development; stricter requirements have been imposed on the industrial production of ester products. The traditional esterification reaction using concentrated sulfuric acid as strong acid catalyst leads to such issues as strong corrosiveness, difficult treatment of waste liquor, and many side reactions [2-3]. Many kinds of heteropoly acids studied later as catalysts also result in such problems as low catalytic efficiency, volatility, and difficulty in recovery [4]. However, the traditional supported acid catalysts have such problems as low selectivity, low catalytic activity and harsh reaction conditions [5]. In recent years, functionalized ionic liquids as new solvents and catalysts have met the requirements of "green

chemistry" and are easy to recover. These ionic liquids are widely used in catalytic reactions, separation engineering and electrochemistry, and because of their unique catalytic properties, they have attracted increasing attention [6]."

We ask you to consider re-wording the abstract, introduction and conclusions to take this into account.

Response: Thanks associate editor for pointing this out. We have rewritten the abstract, introduction and conclusions and altered carefully, and also added some references [2], [3], [6], [7], [8], [9], [10], [12] as follows:

Summary

Four kinds of functional ionic liquids ($[\text{C}_3\text{SO}_3\text{Hnmp}]\text{HSO}_4$, $[\text{C}_3\text{SO}_3\text{Hnmp}]\text{H}_2\text{PO}_4$, $[\text{C}_3\text{SO}_3\text{Hnmp}]\text{CH}_3\text{SO}_3\text{H}$, $[\text{C}_3\text{SO}_3\text{Hnmp}]\text{C}_6\text{H}_6\text{SO}_3\text{H}$) were prepared and the catalytic activity of these ILs during esterification of carboxylic acids (formic acid, acetic acid, propionic acid, butyric acid) with alcohols were investigated. The results indicated that the ionic liquid ($[\text{C}_3\text{SO}_3\text{Hnmp}]\text{HSO}_4$) exhibited an optimal catalytic performance. And then the IL ($[\text{C}_3\text{SO}_3\text{Hnmp}]\text{HSO}_4$) was immobilized to the silica gel. The immobilized ionic liquid performed more excellent catalytic activity than the unsupported $[\text{C}_3\text{SO}_3\text{Hnmp}]\text{HSO}_4$. The effects of reaction temperature, reaction time, molar ratio of acid to alcohol and catalyst dosage were investigated. The response surface methodology (RSM) based on the Box-Behnken design (BBD) was utilized to explore the best reaction condition of different experimental variables. Accordingly, a high n-butyl butyrate yield of 97.10% under the deduced optimal reaction conditions was obtained, in good agreement with experimental results and that predicted by the BBD model. The immobilized ionic liquid $[\text{C}_3\text{SO}_3\text{Hnmp}]\text{HSO}_4$ maintained high catalytic activity after 5 cycles.

Introduction

As an important fine organic chemicals product, carboxylic esters have been widely used in perfume, coating, pharmaceutical intermediates, cosmetics, tobacco and other fields [1]. The main methods to obtain carboxylic esters are extraction of natural substances and chemical synthesis [2-3]. The method of extracting carboxylic ester from natural substance has the disadvantages of complex process, low purity and high

production cost. Therefore, the chemical synthesis method was extensively used to prepare carboxylic esters. And the esterification of alcohols and carboxylic acids became fundamental and important reactions in chemical synthesis. Conventionally, chemical syntheses of carboxylic esters mostly invoke homogeneous catalysts, such as sulphuric acid, p-toluene sulfonic acid, and phosphoric acid etc. And using these high concentrated strong acids as catalyst leads to several problems such as equipment corrosion, environmental pollution, undesirable side reactions, and catalyst unrecyclable [4-5]. To avoid these weaknesses, solid super acids, such as heteropoly acids, strong-acid ion exchange resin, zeolites, and enzymes have been employed for esterification [6-9]. However, acid catalysts also have some problems such as low selectivity and catalytic activity, easy deactivation, formidable separation and unrecyclable and so on [10-11].

Ionic liquids (ILs), as new environmental benign catalysts, which were used to replace conventional catalyst, are widely used in various organic synthesis, separation engineering and electrochemistry owing to their unique characteristics such as adjustable properties, designable structure, low melting point, high thermal stability, negligible volatility, recyclability, and reusability [12-13]. With further research, the immobilized ionic liquids, which were prepared by grafting ionic liquid onto silica gel, molecular sieve, mesoporous nanomaterials and other carriers, can improve the stability and higher catalytic activity in various chemical reactions such as Michael addition, alkylation, epoxidation, Friedle-Crafts, and Heck catalytic hydrogenation [14-17]. Herein, we synthesis four kinds of functional ionic liquids (1-(3-sulfopropyl)-1-methylpyrrolidone sulfate $[C_3SO_3Hnmp]HSO_4$, 1-(3-sulfopropyl)-1-methylpyrrolidone phosphate $[C_3SO_3Hnmp]H_2PO_4$, 1-(3-sulfopropyl)-1-methylpyrrolidone p-toluene sulfonate $[C_3SO_3Hnmp]CH_3SO_3H$, and 1-(3-sulfopropyl)-1-methylpyrrolidone methyl sulfonate $[C_3SO_3Hnmp]C_6H_5SO_3H$), and were used to catalyze the synthesis of n-butyl carboxylate. The ionic liquid $[C_3SO_3Hnmp]HSO_4$ was immobilized onto silica gel, and the immobilized ionic liquid $[C_3SO_3Hnmp]HSO_4$ was applied to the catalytic synthesis of n-butyl butyrate for the first time. The relevant reaction conditions

obtained during the synthesis of n-butyl butyrate were optimized using RSM. Moreover, the immobilized ionic liquid was reused by a simple separating process, and the catalytic activity of reused immobilized ionic liquid was also investigated.

Conclusion

In this paper, a series of Brønsted acidic ionic liquids were synthesized and exploited to catalyse esterification of butyric acid and butyl alcohol. The ionic liquid $[\text{C}_3\text{SO}_3\text{Hnmp}]\text{HSO}_4$ has the best catalytic activity and the highest yield of n-butyl butyrate. So the ionic liquid $[\text{C}_3\text{SO}_3\text{Hnmp}]\text{HSO}_4$ was selected to be immobilized using the sol-gel method. With the optimal esterification conditions: a molar ratio of acid to alcohol of 1.2:1, a reaction temperature of 110°C, and a catalyst dosage of 5% of n-butanol, a maximum n-butyl butyrate yield of 97.10% was obtained catalysed by the immobilized ionic liquid $[\text{C}_3\text{SO}_3\text{Hnmp}]\text{HSO}_4$. This result is consistent with the prediction of the response surface optimization based on the BBD model. Using immobilized ionic liquid $[\text{C}_3\text{SO}_3\text{Hnmp}]\text{HSO}_4$ as catalyst can avoid the mutual dissolution of the catalyst and the solvent, and provide a larger specific surface area, which leads to a significant increasing of the yield of n-butyl butyrate. The yield of n-butyl butyrate remained above 96% after reusing the supported ionic liquid $[\text{C}_3\text{SO}_3\text{Hnmp}]\text{HSO}_4$ 5 times, demonstrating good reusability.

The added references:

- [2] J. F. Qian, H. X. Shi, Z. Yun. 2010 Preparation of biodiesel from *Jatropha curcas* L. oil produced by two-phase solvent extraction. *Bioresource Technology*. 101, 7025-7031. (doi: 10.1016/j.biortech.2010.04.018)
- [3] Son Dinh Le, Shun Nishimura, Kohki Ebitani. 2019 Direct esterification of succinic acid with phenol using zeolite beta catalyst. *Catalysis Communications*. 122, 20-23. (doi: 10.1016/j.catcom.2019.01.006)
- [6] J. Zhao, H.Y. Guan, W. Shi, M.X. Cheng, S.W. Li. 2012 A Brønsted–Lewis-surfactant-combined heteropolyacid as an environmental benign catalyst for esterification reaction. *Catalysis Communications*, 20, 103-106. (doi: 10.1016/j.catcom.2012.01.014)

- [7] Kuzminska, Maryna, Backov, Rénal, Gaigneaux, M. Eric. 2015 Behavior of cation-exchange resins employed as heterogeneous catalysts for esterification of oleic acid with trimethylolpropane. *Applied Catalysis A General*. 504, 11-16. (doi: 10.1016/j.apcata.2014.12.043)
- [8] N. Gokulakrishnan, A. Pandurangan, P.K. Sinha. 2007 Esterification of acetic acid with propanol isomers under autogeneous pressure: A catalytic activity study of Al-MCM-41 molecular sieves. *Journal of Molecular Catalysis A Chemical*. 263, 55-61. (doi: 10.1016/j.molcata.2006.08.005)
- [9] E. H. Ahmed, T. Raghavendra, D. Madamwar. 2010 An alkaline lipase from organic solvent tolerant *Acinetobacter* sp. EH28: Application for ethyl caprylate synthesis. *Bioresource Technology*. 101, 3628-3634. (doi: 10.1016/j.biortech.2009.12.107).
- [10] J. M. Marchetti, A. F. Errazu. 2008 Comparison of different heterogeneous catalysts and different alcohols for the esterification reaction of oleic acid. *Fuel*. 87, 3477-3480. (doi: 10.1016/j.fuel.2008.05.011)
- [12] R. Hagiwara, Y. Ito. 2000 ChemInform abstract: Room temperature ionic liquids of alkylimidazolium cations and fluoroanions. *Journal of Fluorine Chemistry*. 105, 221-227. (doi: 10.1002/chin.200049255)

Response to comments of reviewers 1

1. Page 1, lines 27-31: *The sentence is very difficult to be understood. Which is the subject of the article needs to be clearly explained. Too tedious in current form.*

Response: Thank the reviewer for pointing this out, and the suggestion is of great importance. We have rewritten the abstract seriously, and the revised abstract is as follows: (Page 1, lines 3-12)

Four kinds of functional ionic liquids ($[\text{C}_3\text{SO}_3\text{Hnmp}]\text{HSO}_4$, $[\text{C}_3\text{SO}_3\text{Hnmp}]\text{H}_2\text{PO}_4$, $[\text{C}_3\text{SO}_3\text{Hnmp}]\text{CH}_3\text{SO}_3\text{H}$, $[\text{C}_3\text{SO}_3\text{Hnmp}]\text{C}_6\text{H}_6\text{SO}_3\text{H}$) were prepared and the catalytic activity of these ILs during esterification of carboxylic acids (formic acid, acetic acid, propionic acid, butyric acid) with alcohols were investigated. The results indicated that the ionic liquid $[\text{C}_3\text{SO}_3\text{Hnmp}]\text{HSO}_4$ exhibited an optimal catalytic performance. And then ionic liquid ($[\text{C}_3\text{SO}_3\text{Hnmp}]\text{HSO}_4$) was immobilized to the silica gel. The immobilized ionic liquid performed better catalytic activity than the unsupported $[\text{C}_3\text{SO}_3\text{Hnmp}]\text{HSO}_4$. The effects of reaction temperature, reaction time, molar ratio of acid to alcohol and catalyst dosage were investigated. The response surface methodology (RSM) based on the Box-Behnken design (BBD) was utilized to explore the best reaction condition of different experimental variables. Accordingly, a high n-butyl butyrate yield of 97.21% under the deduced optimal reaction conditions was obtained, in good agreement with experimental results and that predicted by the BBD model. The immobilized ionic liquid $[\text{C}_3\text{SO}_3\text{Hnmp}]\text{HSO}_4$ maintained high catalytic activity after 5 cycles.

2. Page 1, lines 42-43: *please replace “studied” for “were studied”.*

Response: Thanks for the reviewer's careful work. We are very sorry for our incorrect writing, we have checked and corrected.

3. Page 2, lines 23: *please replace “three flasks” to “three-necked flask”. Similar mistakes should be carefully checked.*

Response: Thanks for reviewer's careful work. We have corrected all “three flasks” to “three-necked flask”. (Page 2, lines 29-30)

4. Page 2, lines 21-30: *How do you determine the purity of the products? The differences of the yields were not large (83%-96.8%), but the accuracy of the yield has a great influence on the establishment of the model.*

Response: Thanks for reviewer pointing this out. Just as the reviewer's suggestion, the purity of the products has a great influence on the accuracy of the yield, and also has an influence on the establishment of the model. In our work, we have noticed this problem, and we have purified the product. The purity of the refined product was analyzed by gas chromatography. The purification process is as follows:

After the reaction, the product (n-butyl butyrate as an example) can be separated directly from the reaction system. After separated, dropping sodium bicarbonate into the product until the acid base of product is neutral, and then the product was washed by distilled water. Then adding saturated calcium chloride solution and saturated magnesium sulfate solution to the product successively to remove ethanol, other possible substances and water. Finally, the product was vacuum dried and analyzed by gas chromatography. The gas chromatogram showed that the purity of the product was almost 99%.

Fig. 1 Gas chromatography of product (n-butyl butyrate)

5. Page 8, lines 46-50: *Is any catalyst weight loss detected during the recycling processes?*

Response: Thank the reviewer for serious work. In our work, the catalyst has a small amount of inevitable loss in the process of recycling. When the catalyst was reused for 5 times, there was a nearly 4% mass loss of catalyst.

Response to comments of reviewers 2:

1. The manuscript (ID: RSOS-190166), titled “Catalytic Synthesis of N-Butyl Carboxylate with Immobilized Ionic Liquid Based on RSM Optimization” details a catalytic methodology towards the synthesis of several esters. The reaction scope was evaluated as well as several other aspects such as reaction temperature, reaction time, reutilization of the catalyst system, etc... The results are interesting and probably will interest several researches. However, in my opinion the authors should rewrite the abstract, introduction and conclusions because I detect plagiarism in those parts. I do think that experimental section and/or discussion can have similar descriptions but in the other parts the authors should be innovative.

Response: Thanks for reviewer’s suggestion. We are very sorry that some of the descriptions in our paper are similar to other literatures. We have rewritten the abstract, introduction and conclusions and altered carefully, and also added some references [2], [3], [6], [7], [8], [9], [10], [12] as follows:

Summary

Four kinds of functional ionic liquids ($[\text{C}_3\text{SO}_3\text{Hnmp}]\text{HSO}_4$, $[\text{C}_3\text{SO}_3\text{Hnmp}]\text{H}_2\text{PO}_4$, $[\text{C}_3\text{SO}_3\text{Hnmp}]\text{CH}_3\text{SO}_3\text{H}$, $[\text{C}_3\text{SO}_3\text{Hnmp}]\text{C}_6\text{H}_6\text{SO}_3\text{H}$) were prepared and the catalytic activity of these ILs during esterification of carboxylic acids (formic acid, acetic acid, propionic acid, butyric acid) with alcohols were investigated. The results indicated that the ionic liquid ($[\text{C}_3\text{SO}_3\text{Hnmp}]\text{HSO}_4$) exhibited an optimal catalytic performance. And then the IL ($[\text{C}_3\text{SO}_3\text{Hnmp}]\text{HSO}_4$) was immobilized to the silica gel. The immobilized ionic liquid performed more excellent catalytic activity than the unsupported $[\text{C}_3\text{SO}_3\text{Hnmp}]\text{HSO}_4$. The effects of reaction temperature, reaction time, molar ratio of acid to alcohol and catalyst dosage were investigated. The response surface methodology (RSM) based on the Box-Behnken design (BBD) was utilized to explore the best reaction condition of different experimental variables. Accordingly, a high n-butyl butyrate yield of 97.10% under the deduced optimal reaction conditions was obtained, in good agreement with experimental results and that predicted by the BBD model. The immobilized ionic liquid $[\text{C}_3\text{SO}_3\text{Hnmp}]\text{HSO}_4$ maintained high

catalytic activity after 5 cycles.

Introduction

As an important fine organic chemicals product, carboxylic esters have been widely used in perfume, coating, pharmaceutical intermediates, cosmetics, tobacco and other fields [1]. The main methods to obtain carboxylic esters are extraction of natural substances and chemical synthesis [2-3]. The method of extracting carboxylic ester from natural substance has the disadvantages of complex process, low purity and high production cost. Therefore, the chemical synthesis method was extensively used to prepare carboxylic esters. And the esterification of alcohols and carboxylic acids became fundamental and important reactions in chemical synthesis. Conventionally, chemical syntheses of carboxylic esters mostly invoke homogeneous catalysts, such as sulphuric acid, p-toluene sulfonic acid, and phosphoric acid etc. And using these high concentrated strong acids as catalyst leads to several problems such as equipment corrosion, environmental pollution, undesirable side reactions, and catalyst unrecyclable [4-5]. To avoid these weaknesses, solid super acids, such as heteropoly acids, strong-acid ion exchange resin, zeolites, and enzymes have been employed for esterification [6-9]. However, acid catalysts also have some problems such as low selectivity and catalytic activity, easy deactivation, formidable separation and unrecyclable and so on [10-11].

Ionic liquids (ILs), as new environmental benign catalysts, which were used to replace conventional catalyst, are widely used in various organic synthesis, separation engineering and electrochemistry owing to their unique characteristics such as adjustable properties, designable structure, low melting point, high thermal stability, negligible volatility, recyclability, and reusability [12-13]. With further research, the immobilized ionic liquids, which were prepared by grafting ionic liquid onto silica gel, molecular sieve, mesoporous nanomaterials and other carriers, can improve the stability and higher catalytic activity in various chemical reactions such as Michael addition, alkylation, epoxidation, Friedle-Crafts, and Heck catalytic hydrogenation [14-17]. Herein, we synthesis four kinds of functional ionic liquids (1-(3-sulfopropyl)-1-methylpyrrolidone sulfate $[C_3SO_3Hnmp]HSO_4$,

1-(3-sulfopropyl)-1-methylpyrrolidone phosphate $[C_3SO_3Hnmp]H_2PO_4$
1-(3-sulfopropyl)-1-methylpyrrolidone p-toluene sulfonate $[C_3SO_3Hnmp]CH_3SO_3H$,
and 1-(3-sulfopropyl)-1-methylpyrrolidone methyl sulfonate
 $[C_3SO_3Hnmp]C_6H_5SO_3H$), and were used to catalyze the synthesis of n-butyl
carboxylate. The ionic liquid $[C_3SO_3Hnmp]HSO_4$ was immobilized onto silica gel,
and the immobilized ionic liquid $[C_3SO_3Hnmp]HSO_4$ was applied to the catalytic
synthesis of n-butyl butyrate for the first time. The relevant reaction conditions
obtained during the synthesis of n-butyl butyrate were optimized using RSM.
Moreover, the immobilized ionic liquid was reused by a simple separating process,
and the catalytic activity of reused immobilized ionic liquid was also investigated.

Conclusion

In this paper, a series of Brønsted acidic ionic liquids were synthesized and exploited
to catalyze esterification of butyric acid and butyl alcohol. The ionic liquid
 $[C_3SO_3Hnmp]HSO_4$ has the best catalytic activity and the highest yield of n-butyl
butyrate. So the ionic liquid $[C_3SO_3Hnmp]HSO_4$ was selected to be immobilized
using the sol-gel method. With the optimal esterification conditions: a molar ratio of
acid to alcohol of 1.2:1, a reaction temperature of 110°C, and a catalyst dosage of 5%
of n-butanol, a maximum n-butyl butyrate yield of 97.10% was obtained catalysed by
the immobilized ionic liquid $[C_3SO_3Hnmp]HSO_4$. This result is consistent with the
prediction of the response surface optimization based on the BBD model. Using
immobilized ionic liquid $[C_3SO_3Hnmp]HSO_4$ as catalyst can avoid the mutual
dissolution of the catalyst and the solvent, and provide a larger specific surface area,
which leads to a significant increasing of the yield of n-butyl butyrate. The yield of
n-butyl butyrate remained above 96% after reusing the supported ionic liquid
 $[C_3SO_3Hnmp]HSO_4$ 5 times, demonstrating good reusability.

The added references:

- [2] J. F. Qian, H. X. Shi, Z. Yun. 2010 Preparation of biodiesel from *Jatropha curcas* L.
oil produced by two-phase solvent extraction. *Bioresource Technology*. 101,
7025-7031. (doi: 10.1016/j.biortech.2010.04.018)

- [3] Son Dinh Le, Shun Nishimura, Kohki Ebitani. 2019 Direct esterification of succinic acid with phenol using zeolite beta catalyst. *Catalysis Communications*. 122, 20-23. (doi: 10.1016/j.catcom.2019.01.006)
- [6] J. Zhao, H.Y. Guan, W. Shi, M.X. Cheng, S.W. Li. 2012 A Brønsted–Lewis-surfactant-combined heteropolyacid as an environmental benign catalyst for esterification reaction. *Catalysis Communications*, 20, 103-106. (doi: 10.1016/j.catcom.2012.01.014)
- [7] Kuzminska, Maryna, Backov, Rénal, Gaigneaux, M. Eric. 2015 Behavior of cation-exchange resins employed as heterogeneous catalysts for esterification of oleic acid with trimethylolpropane. *Applied Catalysis A General*. 504, 11-16. (doi: 10.1016/j.apcata.2014.12.043)
- [8] N. Gokulakrishnan, A. Pandurangan, P.K. Sinha. 2007 Esterification of acetic acid with propanol isomers under autogeneous pressure: A catalytic activity study of Al-MCM-41 molecular sieves. *Journal of Molecular. Catalysis A Chemical*. 263, 55-61. (doi: 10.1016/j.molcata.2006.08.005)
- [9] E. H. Ahmed, T. Raghavendra, D. Madamwar. 2010 An alkaline lipase from organic solvent tolerant *Acinetobacter* sp. EH28: Application for ethyl caprylate synthesis. *Bioresource Technology*. 101, 3628-3634. (doi: 10.1016/j.biortech.2009.12.107).
- [10] J. M. Marchetti, A. F. Errazu. 2008 Comparison of different heterogeneous catalysts and different alcohols for the esterification reaction of oleic acid. *Fuel*. 87, 3477-3480. (doi: 10.1016/j.fuel.2008.05.011)
- [12] R. Hagiwara, Y. Ito. 2000 ChemInform abstract: Room temperature ionic liquids of alkylimidazolium cations and fluoroanions. *Journal of Fluorine Chemistry*. 105, 221-227. (doi: 10.1002/chin.200049255)

We tried our best to improve the manuscript and made some changes in the manuscript. These changes will not influence the content and framework of the paper. And we marked the changes in red in revised paper. We appreciate for Reviewers' warm work earnestly, and hope that the correction will meet with approval. Once

again, thank you and all the reviewers for the kind advice.

Sincerely yours

Ran Liu

21 May 2019

Appendix B

Response to editor

Dear editor

Thank you very much for your patience and careful help. Through your reply letter, we understand how to properly arrange our figures. We have sorted out all the data and divided them into main document, supplementary material (ESM files), and Dryad reasonably, and there is no duplication of data in each part.

Response to comments of editor and reviewers

Dear editor and reviewers:

Thank you for your comments concerning our manuscript entitled “Catalytic Synthesis of N-Butyl Carboxylate with Immobilized Ionic Liquid Based on RSM” (ID: RSOS-190166.R1), We have studied comments given by reviewers 2 carefully and have made correction which we hope meet with approval. Revised portion are marked in red in the paper. The main corrections in the paper and the responds to comments are as flowing:

Response to comments of reviewers 2:

1. The authors improved the manuscript and it is much better now. However, they should decide if the figures will be presented in the manuscript or in the supplementary data. In my opinion they should put less in the manuscript. Nevertheless, they should not repeat the same figures in both documents.

Response: Thanks for reviewer’s suggestion. We carefully considered the pictures in our article and decided to put 5 pictures into the supplementary materials. There are no same figures in both documents. These changes will not influence the content and framework of the paper.

The following 5 pictures are put into the supplementary materials.

Fig. S-1 The 3D diagram of effect of B and D on yield

Fig. S-2 The 3D diagram of effect of B and C on yield

Fig. S-3 The 3D diagram of effect of A and C on yield

Fig. S-4 The 3D diagram of effect of A and D on yield

Fig. S-5 The 3D diagram of effect of A and B on yield

Fig. S-6 The 3D diagram of effect of C and D on yield

We appreciate for Reviewers' warm work earnestly, and hope that the correction will meet with approval. Once again, thank you and all the reviewers for the kind advice.

Sincerely yours

Ran Liu
10 July 2019